# Flavonoids from *Acer okamotoanum* Inhibit Adipocyte Differentiation and Promote Lipolysis in the 3T3-L1 Cells

**DOI:** 10.3390/molecules25081920

**Published:** 2020-04-21

**Authors:** Ji Hyun Kim, Sanghyun Lee, Eun Ju Cho

**Affiliations:** 1Department of Food Science and Nutrition & Kimchi Research Institute, Pusan National University, Busan 46241, Korea; kjjjjhh11@naver.com; 2Department of Plant Science and Technology, Chung-Ang University, Anseong 17546, Korea; slee@cau.ac.kr

**Keywords:** *Acer okamotoanum*, afzelin, isoquercitrin, obesity, quercitrin

## Abstract

Flavonoids, quercitrin, isoquercitrin (IQ), and afzelin, were isolated from ethyl acetate fraction of *Acer okamotoanum*. We investigated anti-obesity effects and mechanisms of three flavonoids from *A. okamotoanum* in the differentiated 3T3-L1 cells. The differentiated 3T3-L1 cells increased triglyceride (TG) contents, compared with non-differentiated normal group. However, treatments of three flavonoids from *A. okamotoanum* decreased TG contents without cytotoxicity. In addition, they showed significant down-regulation of several adipogenic transcription factors, such as γ-cytidine-cytidine-adenosine-adenosine-thymidine/enhancer binding protein -α, -β, and peroxisome proliferator-activated receptor-γ, compared with non-treated control group. Furthermore, treatment of the flavonoids inhibited expressions of lipogenesis-related proteins including fatty acid synthase, adipocyte protein 2, and glucose transporter 4. Moreover, IQ-treated group showed significant up-regulation of lipolysis-related proteins such as adipose triglyceride lipase and hormone-sensitive lipase. In addition, flavonoids significantly activated 5′-adenosine monophosphate-activated protein kinase (AMPK) compared to control group. In particular, IQ showed higher inhibition of TG accumulation by regulation of pathways related with both adipogenesis and lipolysis, than other flavonoids. The present results indicated that three flavonoids of *A. okamotoanum* showed anti-obesity activity by regulation of adipocyte differentiation, lipolysis, and AMPK signaling, suggesting as an anti-obesity functional agents.

## 1. Introduction

Obesity has been associated with various degenerative diseases, such as cardiovascular disease, type 2 diabetes mellitus, atherosclerosis, and Alzheimer’s disease [1,2]. The environmental factors such as consumption of food and lifestyle play the most important role among other causes of obesity, in particular, excessive energy intake leads to increase of the adipose tissue and accumulation of triglyceride (TG) [3]. Adipose tissue acts energy metabolism by secreting adipokines, but its excessive expansion is closely related to the development of obesity [4]. The differentiation of preadipocytes (adipogenesis) results in intracellular lipid accumulation in adipose tissue of patients with obesity [5].

Adipogenesis is the process to become adipocytes mature from preadipocytes, and it is regulated by adipogenic key transcription factors including family of CCAAT/enhancer-binding proteins (C/EBPs) and peroxisome proliferator-activated receptors (PPARs) [6]. During adipogenesis process, adipogenic key transcription factors enhance the lipogenic enzymes such as fatty acid synthase (FAS) and adipocytes protein 2 (aP2) [7,8]. Lipolysis is the process that hydrolyzes TG and diglyceride by activation of hydrolyzing enzymes including hormone-sensitive lipase (HSL) and adipose triglyceride lipase (ATGL), resulting in the production of fatty acids and glycerol [9]. In addition, lipolysis is decreased by over-expression of adipogenic key transcription factors in the adipogenesis [9]. Moreover, phosphorylation of 5′-adenosine monophosphate-activated protein kinase (AMPK) is important for progression of obesity [10]. Therefore, many researchers are focused on treatment of obesity by regulation of AMPK pathway [10,11]. Recently, anti-obesity agents are popular and attracted much attention, but they have several side effects including diarrhea, headache, and gastrointestinal discomfort [12]. Therefore, finding of natural products without side effects for anti-obesity agents by inhibition of adipogenesis and promotion of lipolysis is one of strategies to prevent and treat obesity.

*Acer okamotoanum* is widely distributed in Ulleung-do, Republic of Korea, as one variety of Korean endemic species [13]. *A. okamotoanum* has been reported to exert some biological activities such as anti-oxidant, anti-hypertension and immune improvement effect [14,15,16]. *A. okamotoanum* contains biological compounds such as cleomiscosin A and C, gallic acid, and β-amyrin [17]. We previously isolated flavonoids such as quercitrin (QU; quercetin-3-rhamoside), isoquercitrin (IQ; quercetin-3-glucoside), and afzelin (AF; kaempferol-3-rhamoside) from ethyl acetate (EtOAc) fraction of *A. okamotoanum* [18]. Therefore, in the present study, we investigated anti-obesity effects of three flavonoids from *A. okamotoanum* including QU, IO, and AF in the differentiated 3T3-L1 cells. In addition, molecular mechanisms related to anti-obesity effects of flavonoids from *A. okamotoanum* on adipogenesis and lipolysis was also observed.

## 2. Results

### 2.1. Effects of Flavonoids from A. okamotoanum on Differentiation of Preadipocytes and Lipid Accumulation

We investigated cytotoxicity of flavonoids from *A. okamotoanum* at the doses (1–10 µg/mL) in the 3T3-L1 adipocytes. As shown in Figure 1, treatment of three flavonoids at concentrations up to 10 µg/mL had no significant cytotoxicity on 3T3-L1 adipocytes, compared with non-treated normal group. Therefore, we used flavonoids at the concentration up to 10 µg/mL in this study.

To evaluate the effects of flavonoids from *A. okamotoanum* on differentiation of preadipocytes and lipid accumulation, we conducted Oil Red O staining, and then visualized cell morphology by light microscopy (Figure 2). The control group showed cell differentiation and lipid droplets induced by treatment of 3-isobutyl-1-methylxanthine, dexamethasone, and insulin (MDI), compared with normal group. However, treatment of three flavonoids such as QU, IQ, and AF at 10 µg/mL inhibited differentiation of preadipocytes and lipid droplets production, compared with control group. In particular, IQ inhibited more effectively differentiation and lipid droplets among other flavonoids.

We also measured intracellular TG contents by Oil Red O quantification (Figure 3). Non-differentiated normal group showed 7.64% TG contents, while control group showed 100.00% of TG contents. On the other hand, TG content of the flavonoids-treated groups such as QU, IQ, and AF at 10 µg/mL is decreased to 83.30%, 18.88%, and 90.17%, respectively. In particular, IQ-treated group inhibited accumulation of TG the most effectively.

### 2.2. Effects of Flavonoids from A. okamotoanum on Expressions of Adipogenic Key Transcription Factors

To confirm the effects of flavonoids such as QU, IQ, and AF on adipogenic key transcription factors, we measured the protein expressions of C/EBPs family such as C/EBPα, C/EBPβ, and PPARs family including PPARγ. As shown in Figure 4, MDI-stimulated control group cells significantly increased these adipogenic key transcription factors such as C/EBPα, C/EBPβ, and PPARγ. In the C/EBPs family expressions, IQ- and AF-treated group showed significant down-regulation of C/EBPα and C/EBPβ levels, compared with control group. In PPARs expression, three flavonoids-treated group significantly decreased PPARγ level compared with control group. Especially, the treatment of IQ inhibited protein expressions of both C/EBPs and PPARs family.

### 2.3. Effects of Flavonoids from A. okamotoanum on Lipogenesis-Related Protein Expressions

To examine the effects of flavonoids such as QU, IQ, and AF on lipogenesis-related protein expressions, we carried out the measurement of protein expressions such as FAS, aP2, and glucose transporter 4 (GLUT4). In our results (Figure 5), the stimulation of MDI significantly increased the lipogenesis-related protein expressions such as FAS, aP2, and GLUT4. Treatment with 10 µg/mL of three flavonoids including QU, IQ, and AF resulted in significant down-regulation of protein expressions involved in lipogenesis. In addition, treatment with IQ showed higher decrease in protein expressions of FAS and aP2, whereas QU- or IQ-treated group showed down-regulation of GLUT4 protein expression.

### 2.4. Effects of Flavonoids from A. okamotoanum on Lipolysis-Associated Proteins

As shown in Figure 6, we examined the effects of flavonoids including QU, IQ, and AF on protein expressions associated with lipolysis such as ATGL and HSL. The protein expression of ATGL was significantly decreased by MDI stimulation, whereas ATGL expression was increased by treatment of QU or IQ in the differentiated 3T3-L1 cells. In addition, MDI stimulation significantly decreased phosphorylation of HSL, compared with MDI-non-stimulation. However, treatment with IQ significantly increased phosphorylation of HSL in the differentiated 3T3-L1 cells. In particular, IQ significantly up-regulated lipolysis- associated protein expressions, both ATGL and HSL.

### 2.5. Effects of Flavonoids from A. okamotoanum on Activation of AMPK Signaling

We further investigated the effect of flavonoids on AMPK signaling pathway. The expressions of AMPK and ACC phosphorylation were determined by western blotting (Figure 7). The levels of AMPK phosphorylation was significantly lower in differentiated 3T3-L1 cells. On the other hand, treatment with flavonoids including QU and IQ showed significant high levels of AMPK phosphorylation compared with control group. In addition, three flavonoids-treated groups also significantly elevated the levels of ACC phosphorylation, indicating that three flavonoids up-regulated downstream target of AMPK pathway.

## 3. Discussion

Obesity is caused by an excessive accumulation of lipid in adipose tissue. Adipocytes function for lipid homeostasis and energy balance by storing TG or releasing free fatty acids [19]. However, abnormal increase in number and size of differentiated adipocytes from preadipocytes has resulted in obesity by increasing of adipose tissue mass [20]. To prevent obesity, balance of adipogenesis and lipolysis in adipocytes plays important roles. Inhibition of preadipocyte differentiation as well as inductions of lipolysis has been focused on therapeutic strategies for treating obesity [21]. In particular, differentiation of 3T3-L1 preadipocytes induced by MDI media showed the characteristics of mature adipocyte in the growth, metabolism, and lipid accumulation [22]. Therefore, 3T3-L1 preadipocyte is widely used as an obesity-related study.

Recently, various dietary supplements are focused on the treatment of obesity. In particular, safe and effective natural products have been used worldwide as preventive agents for obesity and its related metabolic diseases [23]. *A. okamotoanum* is an endemic natural product and contains various bioactive compounds such as flavonoids [13]. We previously isolated three flavonoid glycosides such as QU, IQ, and AF from EtOAc fraction of *A. okamotoanum* [18]. QU and IQ are quercetin containing rhamnoside and glucoside, respectively, and AF is kaempferol with rhamnoside. However, anti-obesity effect and mechanisms of these flavonoids from *A. okamotoanum* including QU, IQ, and AF have not been investigated. In this study, we investigated the effects of flavonoids from *A. okamotoanum* on lipid accumulation, adipogenesis, lipolysis, and AMPK signaling under differentiated 3T3-L1 cells. Differentiated 3T3-L1 cells induced by MDI media showed significant increase of lipid accumulation, compared with non-differentiated 3T3-L1 cells. However, treatment of three flavonoids significantly decreased contents of TG. It indicated the promising role as an anti-obesity agent by inhibition of lipid accumulation.

To determine the concentration of flavonoids, the cytotoxicity assay was carried out. Three flavonoids used in the present study had no significant cytotoxicity up to concentrations of 10 μg/mL (Figure 1). We previously investigated that three flavonoids had no cytotoxicity up to concentrations of 50 μg/mL in 3T3-L1 cells (data not shown). In addition, previous studies examined anti-adipogenic effects of IQ at concentrations of 5–100 μM in 3T3-L1 cells, indicating no cytotoxicity under higher concentrations used in this study [24,25]. Therefore, in this study, inhibition of triglyceride accumulation by flavonoids was probably related to regulation of adipogenesis from 3T3-L1 preadipocytes to differentiated 3T3-L1 cells without cytotoxicity. In addition, we designed experimental schedule in measurement of cytotoxicity with reference to previous other studies [26,27].

C/EBPs and PPARs family are essential factors for adipogenesis [6]. Adipogenesis accelerates differentiation of preadipocytes into adipocytes, resulting in intracellular lipid accumulation [5,6]. During early stage of adipogenesis, cAMP agonist (IBMX) and glucocorticoids (dexamethasone) in the MDI directly induce expression of C/EBPβ [28,29]. C/EBPβ induced by MDI activates the expression of two major adipogenic transcription factors such as C/EBPα and PPARγ [30]. C/EBPα is intimately related to PPARγ activity, and also deficiency of C/EBPα in the adipocytes inhibits adipogenesis [28,31]. PPARγ is a nuclear receptor related with differentiation of adipose tissue, and it regulates the transcription of target genes associated with lipid homeostasis [32]. In addition, it is widely known that differentiated 3T3-L1 cells by treatment with MDI activate adipogenic key transcription factors including C/EBPα, C/EBPβ, and PPARγ [29]. In our results, we confirmed that differentiated 3T3-L1 cells showed up-regulation of adipogenic key transcription factors including C/EBPα, C/EBPβ, and PPARγ, whereas treatment with flavonoids showed down-regulation of these adipogenic transcription factors. In particular, IQ-treated group in differentiated 3T3-L1 cells reduced more effectively the expressions of three adipogenic transcription factors including C/EBPα, C/EBPβ, and PPARγ than other flavonoids. It indicated that IQ could inhibit the lipid accumulation by down-regulations of the adipogenic transcription factors.

Adipogenic key transcription factors such as C/EBPs and PPARs family activate the adipocyte specific proteins such as FAS, aP2, and GLUT4. C/EBPα and PPARγ activation is known to regulate the lipid metabolism-related genes such as FAS and aP2 [33]. FAS is one of the lipogenesis associated enzymes to facilitate the synthesis and cytoplasmic storage of massive amounts of TG during differentiation process [7,8]. C/EBPs activate expression of GLUT4, that is required during insulin-dependent glucose uptake [34,35]. Expression of GLUT4 is increased by activation of adipogenic key transcription factor such as C/EBPα in adipocyte differentiation [36]. Previous study reported that decrease of C/EBPα inhibited expression of GLUT, which would decrease glucose transport in adipocytes [36,37]. In our study, treatment of IQ decreased the expression of C/EBPα and GLUT4. Therefore, IQ inhibited the accumulation of TGs by decreasing C/EBPα-activated GLUT4 expression. GLUT4 is a member of glucose transporter in adipose tissue, and involved in the insulin-stimulated glucose uptake [38,39]. On the basis of these evidences, the inhibitory effect of IQ on TG accumulation is related to GLUT4 by regulation of C/EBPα expressions. In addition, differentiated 3T3-L1 cells treated with IQ effectively reduced expressions of aP2 and FAS by down-regulation of adipogenic key transcription factors among other flavonoids.

Lipolysis in the adipose tissue is regulated by ATGL and HSL that promote lipolysis by acting on the lipid droplet [9]. Expression of ATGL initiates lipolysis by cleaving the fatty acid from triacylglycerol, and then HSL catalyzes the hydrolysis of diacylglycerol [40]. Translocation of HSL is regulated by increase of cAMP and activation of protein kinase A, especially regulation of HSL is a major rate-limiting step in adipocytes during lipolysis [40,41]. Therefore, activation of ATGL and HSL are the strategy for treatment of obesity by activation of lipolysis. In this study, QU and IQ significantly up-regulated expression of ATGL. Furthermore, differentiated 3T3-L1 cells treated with IQ significantly elevated ATGL as well as phosphorylation of HSL. These results indicated that QU and IQ promote lipolysis by regulation of ATGL and HSL.

AMPK is a common regulator of lipid metabolism in adipocytes [10,42]. AMPK involves in various lipid metabolisms such as adipogenesis, lipolysis, glucose uptake, fatty acid β-oxidation, and adipokine secretion in the adipose tissue [42]. In the adipogenesis, activation of AMPK can inhibit adipocyte differentiation via reductions of transcription key factors such as PPARγ and C/EBPα [43,44]. It has been reported that activation of AMPK directly down-regulated adipogenic key transcription factors and inhibited adipocyte differentiation [43,44]. In the lipolysis process, activation of AMPK promotes lipolysis by phosphorylation of HSL and ATGL, resulting in fatty acid degradation [10,45]. In addition, AMPK enhances inactivation of ACC by phosphorylation and decrease in the free fatty acid production [11]. ACC is known as reduction of fatty acid synthesis and elongation in adipocytes [46,47]. Therefore, activation of AMPK leads to attenuation of adipogenesis and increase of lipolysis, thereby regulation of AMPK signaling pathway plays important roles for prevention and treatment of obesity [48]. In our study, IQ-treated group significantly increased phosphorylation of both AMPK and ACC among other flavonoids from *A. okamotoanum* in the differentiated 3T3-L1 cells. Therefore, we suggest that IQ effectively suppresses the adiopgenesis and promotes lipolysis via regulation of AMPK pathway, resulting in inhibition of lipid accumulation.

Dietary flavonoids from natural products have anti-obesity effects via regulation of various molecular pathways [49]. Flavonoids are the most abundant polyphenols in natural products, and they can be classified into six groups including flavones, flavonols, flavanones, flavanonols, flavanols, and antocyanin [50]. Dietary flavonoids from natural products exert anti-obesity effects via regulation of various molecular pathways in 3T3-L1 cells [49]. Flavones such as apigenin, balcalein, and lueteolin inhibited lipid accumulation by downregulation of adipogenesis, activation of AMPK, reduction of Akt-C/EBPa-GLUT4 signaling-medicated glucose uptake, and inflammatory responses [35,51,52]. Flavanones including hesperitin and naringin exert antiobesity effects via activating PPAR and up-regulating adiponectin in 3T3-L1 cells [53]. Anthocyanidins isolated from black soybean such as cyanidine, peonidin, and its glucoside reduced preadipocyte differentiation. Flavan-3-ols such as epigalloocatechin gallate in the green tea inhibited adipogenesis via regulation of Wnt/β-catenin pathway in 3T3-L1 cells [49].

In our study, flavonoids from *A. okamotoanum* including QU, IQ, and AF are classified as flavonols. Flavonols including quercetin, kaempferol, and rutin are flavonoids with a ketone group in onions, tomatoes, apples, and berries [50]. Flavonols such as quercetin, kaempferol, and its glycosides inhibited adipogenesis or promoted AMPK signaling in adipocytes [54,55]. In comparison of aglycone and its glycosides, quercetin glucoside showed strong activity in reduction of lipid accumulation and inhibited adipogenic factors such as C/EBP-β, -α, and aP2 than its aglycone, quercetin [54]. Wnt/β-catenin signaling is mandatory in adipogenesis, and it inhibited preadipocyte differentiation by regulation of β-catenin [25]. Lee et al. reported that IQ suppressed the adipogenesis in 3T3-L1 cells via the inhibition of Wnt/β-catenin signaling, regulation of lipid metabolism-related factors such as PPARγ, C/EBPα, SREBP-1, adiponectin, resistin, visfatin, and improvement of insulin resistance [54,56]. In addition, our results indicated the inhibitory effect of lipid accumulation of flavonoids from *A. okamotoanum* including QU, IQ, and AF, in particular IQ, by regulation of adipogenesis, lipolysis, and AMPK signaling. IQ has hydroxyl group at R1 position and glycosylated glucoside at R2 position. We suggest that presence of hydroxyl group at R1 position and glycosylation of glucoside at R2 position are higher anti-obesity effects compared with flavonols non-linked hydroxyl group at R1 and glucoside at R2 such as QU and AF. Previous study demonstrated that extract of *A. okamotoanum* leaf inhibited the anti-adipogenesis via down-regulation of PPARγ and C/EBPα [14]. However, anti-obesity effects and molecular mechanisms of flavonoids isolated from *A. okamotoanum* have not been demonstrated. This study indicated that the flavonoids of *A. okamotoanum,* in particular IQ, exerted anti-obesity effects by regulation of adipogenesis and lipolysis.

## 4. Materials and Methods

### 4.1. Sample Preparation

The isolation and identification of three flavonoids such as QU, IQ, and AF were based on our previous study [18]. In brief, *A. okamotoanum* was obtained from Ulleung-do, Gyeongsangbuk-do, Korea. A voucher specimen has been deposited at the Department of Plant Science and Technology, Chung-Ang University, Anseong, Korea (Voucher No. LEE 2014-04). Aerial parts of *A. okamotoanum* (995.4 g) were extracted with methanol (MeOH) by filtering and evaporation *in vaccum*. EtOAc fraction of *A. okamotoanum* (35.0 g) was partitioned from MeOH extract, and then the fraction of *A. okamotoanum* was isolated by silica gel column chromatography. Three flavonoids were identified as QU, IQ, and AF by ^1^H-NMR, ^13^C-NMR, and MS data. In addition, quantitative analysis of three flavonoids such as QU (48.26 μg/g), IQ (5.84 μg/g), and AF (2.66 μg/g) was determined by HPLC/UV using 0.45 μM syringe filter. These three flavonoids including QU, IQ, and AF were isolated from *A. okamotoanum* of EtOAc fraction [18]. Figure 8 and Table 1 expressed the structures of three flavonoids from *A. okamotoanum* such as QU, IQ, and AF.

### 4.2. Reagents

Fetal bovine serum (FBS), bovine calf serum (BCS), Dulbecco’s modified eagle medium (DMEM), penicillin-streptomycin, and trypsin-EDTA solution were purchased from Welgene (Daegu, Korea). Dexamethasone, 3-isobutyl-1-methylxanthine (IBMX), and insulin were purchased from Sigma Aldrich (St. Louis, MO, USA). 3-(4,5-Dimethyl-2-thiazolyl)-2,5-diphenyl-2H-tetrazolium bromide (MTT) was acquired from Bio Basic (Toronto, ON, Canada), dimethyl sulfoxide (DMSO) was purchased from Bio Pure (Burlington, ON, Canada), and primary and secondary antibodies were purchased from Cell Signaling Technology (Beverly, MA, USA).

### 4.3. Cell Culture and Differentiation

The mouse 3T3-L1 preadipocytes were purchased from American Type Culture Collection (Maryland, OH, USA). The cells were grown in DMEM supplemented with 10% BCS and 1% penicillin-streptomycin, at 37 °C in 5% CO_2_ atmosphere incubator (Thermo Electron Corporation, Milford, MA, USA). The differentiation was induced by 0.5 mM IBMX, 1 μM dexamethasone, and 5 μg/mL insulin in DMEM containing 10% FBS (MDI media) under the presence or absence of flavonoids including QU, IQ, and AF. After 2 days, MDI media was replaced with DMEM supplemented with 10% FBS and 5 μg/mL insulin (insulin media). After during 8 days, the cell culture media was replaced with insulin media 4 times every 2 day.

### 4.4. Cell Viability

The cell viability was evaluated using a MTT assay [57]. The 3T3-L1 preadipocytes were seeded at a density of 1 X 10^5^ cells/mL in the 24 well plate, and incubated for 24 h. Flavonoids such as QU, IQ, and AF were treated in the wells at various concentrations (1, 2.5, 10 µg/mL), respectively, and then incubated at 37 °C for 72 h. Cell culture media was removed, and then MTT solution was added in the wells. The formazan crystals were dissolved in DMSO and the absorbance was measured at 540 nm using microplate reader (Thermo Fisher Scientific, Vantaa, Finland).

### 4.5. Oil Red O Staining

After 8 days of differentiation, differentiated 3T3-L1 cells were washed with phosphate buffered saline (PBS), fixed with 10% formalin for 10 min and washed with PBS and 60% isopropanol. The fixed cells were stained with 0.6% Oil Red O solution for 20 min in the dark, washed 4 times with PBS and 60% isopropanol. The lipid droplets within the differentiated 3T3-L1 cells were visualized and photographed by microscopy. For quantitative analysis under Oil Red O staining was carried out by eluting with 100% isopropanol and measuring absorbance at 500 nm [58].

### 4.6. Western Blot Analysis

The differentiated 3T3-L1 cells were harvested using a cell scraper, lysed with radioimmunoprecipitation assay buffer containing protease inhibitor cocktail at 4 °C for 1 h. The protein concentration was determined using Bio-Rad protein assay (Bio-Rad, Hercules, CA, USA). Equal amounts of extracted proteins were separated on 8–13% sodium dodecyl sulfate-polyacrylamide gel electrophoresis, transferred onto a polyvinylidene fluoride membrane. The each membrane was blocked 5% skim milk at room temperature for 1 h, and then incubated primary antibodies such as β-actin, C/EBPα, C/EBPβ, PPARγ, FAS, ap2, GLUT4, ATGL, phospho-HSL, HSL, phospho-AMPK, AMPK, phospho-acetyl-CoA carboxylase (ACC), and ACC at 4 °C for overnight. The next day, each membrane was washed with PBS-T, and then incubated with secondary antibodies at room temperature for 1 h. The protein bands were activated with enhanced chemiluminescence solution, visualized with Davinch-chemi^TM^ Chemiluminescence Imaging System (Core Bio, Seoul, Korea), and quantified with Image-J program (National Institutes of Health, Bethesda, MD, USA).

### 4.7. Statistical Analysis

Values are expressed as mean ± standard deviation (SD). SPSS software (IBM SPSS Inc., Chicago, IL, USA) was used to perform statistics analysis. The results were compared by one-way analysis of variance (ANOVA) followed by Duncan’s multiple range analysis among the groups using SPSS program. Statistically significance was considered as *P* < 0.05.

## 5. Conclusions

In conclusion, this study demonstrated that three flavonoids from *A. okamotoanum* such as QU, IQ, and AF suppressed adipogenesis by inhibition of adipogenic key transcription factors and lipogenesis-related proteins, resulting in decrease of TG accumulation in differentiated 3T3-L1 cells. In addition, flavonoids from *A. okamotoanum* promoted lipolysis by down-regulation of lipolytic genes expressions. In particular, IQ effectively suppressed adipogenesis and induced lipolysis by regulation of AMPK signaling among other flavonoids of *A. okamotoanum*. The present study suggests that the flavonoids from *A. okamotoanum* can be the promising agents for the prevention and treatment of obesity.

## Figures and Tables

**Figure 1 molecules-25-01920-f001:**
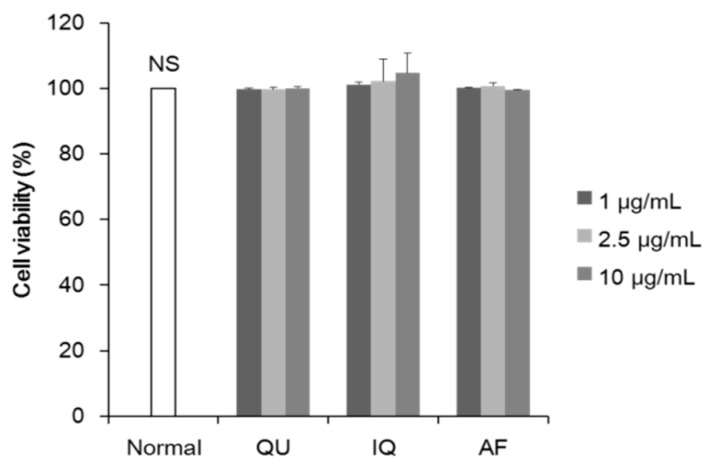
Effects of flavonoids from *Acer okamotoanum* on the cell viability in 3T3-L1 adipocytes. The 3T3-L1 adipocytes were pretreated with various concentrations (1–10 µg/mL) of flavonoids from *A. okamotoanum* for 72 h. Data are expressed as the mean ± standard deviation. NS: Non-significance; QU: Quercitrin; IQ: Isoquercitrin; AF: Afzelin.

**Figure 2 molecules-25-01920-f002:**
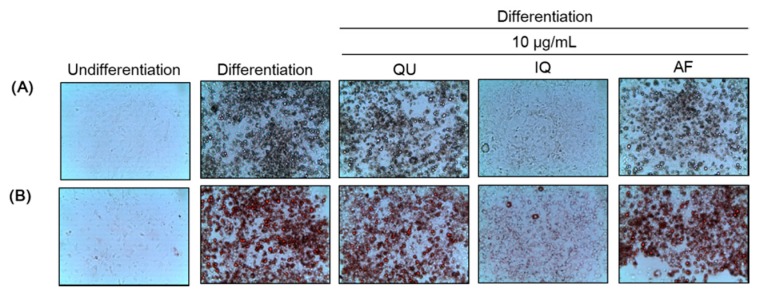
Effects of flavonoids from *Acer okamotoanum* on cell differentiation in differentiated 3T3-L1 cells. Adipocyte differentiation was induced by treatment with 3-isobutyl-1-methylxanthine, dexamethasone, and insulin (MDI) media in the absence or presence of flavonoids from *A. okamotoanum* during 2 days. The MDI media was then replaced with insulin media, and it was changed four times for every 2 days. The cells were confirmed by light microscopy (magnification, ×100) (**A**). Cells were fixed and stained with Oil Red O staining to visualize the lipid droplets by light microscopy (magnification, ×100) (**B**). Normal group indicates non-differentiated cells, whereas control group indicates the differentiated cells by treatment of MDI media. QU: Quercitrin; IQ: Isoquercitrin; AF: Afzelin.

**Figure 3 molecules-25-01920-f003:**
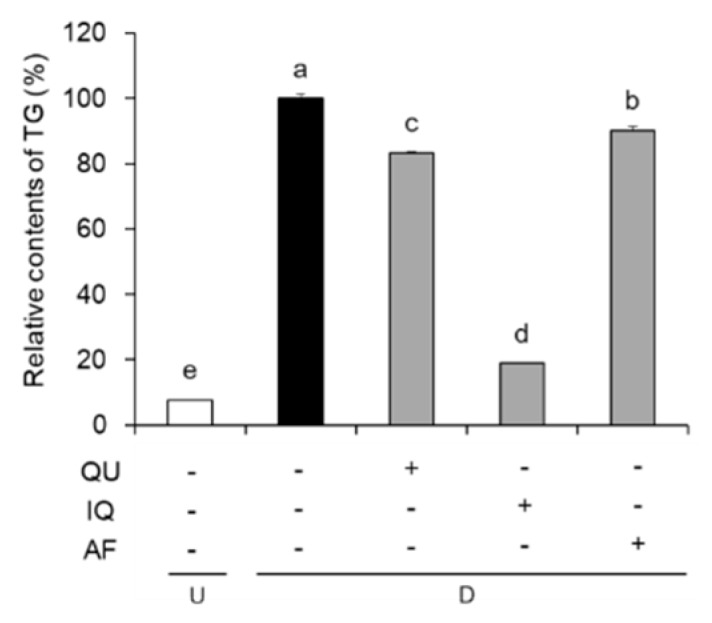
Effects of flavonoids from *Acer okamotoanum* (10 µg/mL) on intracellular triglyceride (TG) accumulation in differentiated 3T3-L1 cells. Adipocyte differentiation was induced by treatment with MDI media in the absence or presence of flavonoids from *A. okamotoanum* during 2 days. The MDI media was then replaced with insulin media, and it was changed four times for every 2 days. Data are expressed as the mean ± standard deviation. ^a^^–e^ Means with different letters indicate significant differences (*P* < 0.05) by Duncan’s multiple range test. Normal group indicates non-differentiated cells, whereas control group indicates the differentiated cells by treatment of MDI media. U: Undifferentiation; D: Differentiation; QU: Quercitrin; IQ: Isoquercitrin; AF: Afzelin.

**Figure 4 molecules-25-01920-f004:**
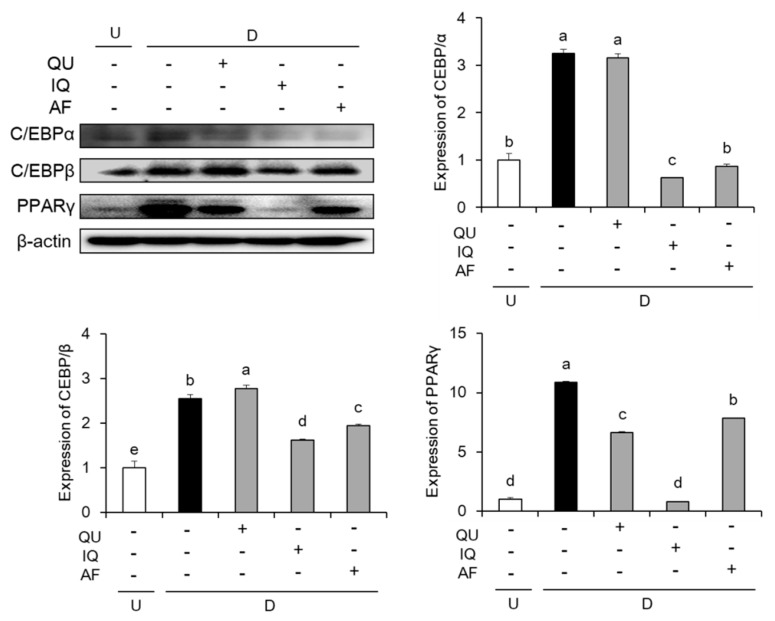
Effects of flavonoids from *Acer okamotoanum* (10 µg/mL) on adipogenic key transcription factors in differentiated 3T3-L1 cells. Adipocyte differentiation was induced by treatment with MDI media in the absence or presence of flavonoids from *A. okamotoanum* during 2 days. The MDI media was then replaced with insulin media, and it was changed four times for every 2 days. Data are expressed as the mean ± standard deviation. ^a^^–e^ Means with different letters indicate significant differences (*P* < 0.05) by Duncan’s multiple range test. β-actin was used as a loading control. Relative expression levels were normalized to the β-actin levels and the normal group. Normal group indicates non-differentiated cells, whereas control group indicates the differentiated cells by treatment of MDI media. U: Undifferentiation; D: Differentiation; QU: Quercitrin; IQ: Isoquercitrin; AF: Afzelin.

**Figure 5 molecules-25-01920-f005:**
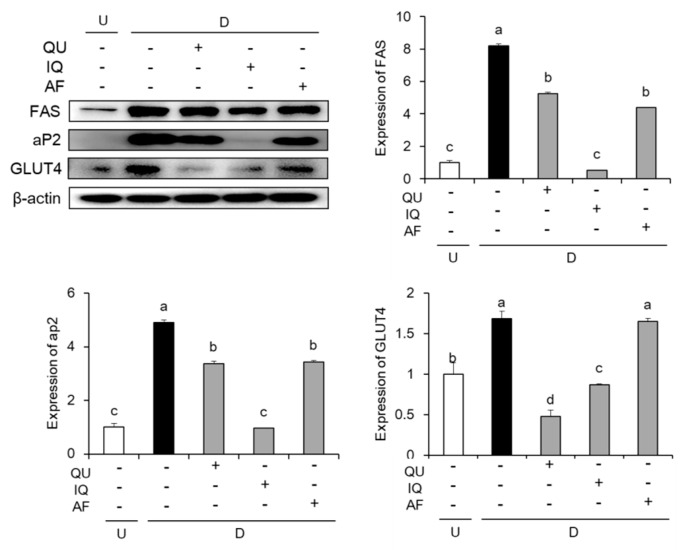
Effects of flavonoids from *Acer okamotoanum* (10 µg/mL) on lipogenesis-related protein expressions in differentiated 3T3-L1 cells. Adipocyte differentiation was induced by treatment with MDI media in the absence or presence of flavonoids from *A. okamotoanum* during 2 days. The MDI media was then replaced with insulin media, and it was changed four times for every 2 days. Data are expressed as the mean ± standard deviation. ^a^^–d^ Means with different letters indicate significant differences (*P* < 0.05) by Duncan’s multiple range test. β-actin was used as a loading control. Relative expression levels were normalized to the β-actin levels and the normal group. Normal group indicates non-differentiated cells, whereas control group indicates the differentiated cells by treatment of MDI media. U: Undifferentiation; D: Differentiation; QU: Quercitrin; IQ: Isoquercitrin; AF: Afzelin.

**Figure 6 molecules-25-01920-f006:**
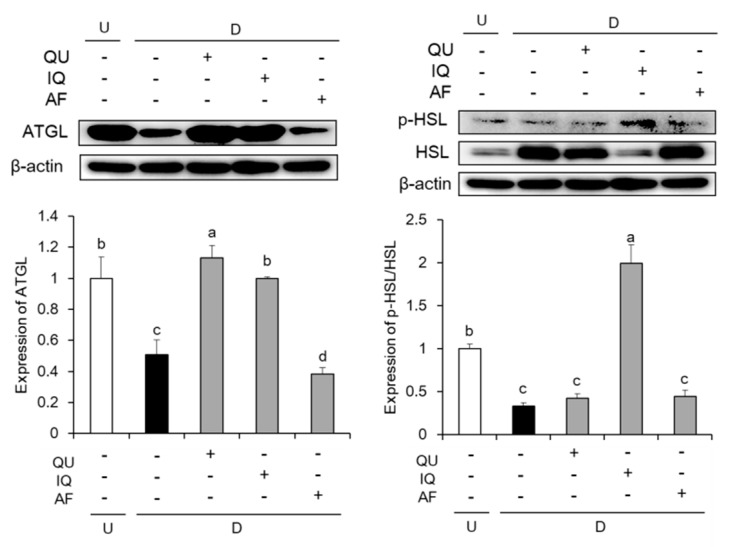
Effects of flavonoids from *Acer okamotoanum* (10 µg/mL) on lipolysis associated protein expressions in differentiated 3T3-L1 cells. Adipocyte differentiation was induced by treatment with MDI media in the absence or presence of flavonoids from *A. okamotoanum* during 2 days. The MDI media was then replaced with insulin media, and it was changed four times for every 2 days. Data are expressed as the mean ± standard deviation. ^a^^–d^ Means with different letters indicate significant differences (*P* < 0.05) by Duncan’s multiple range test. β-actin was used as a loading control. Relative expression levels were normalized to the β-actin levels and the normal group. Normal group indicates non-differentiated cells, whereas control group indicates the differentiated cells by treatment of MDI media. U: Undifferentiation; D: Differentiation; QU: Quercitrin; IQ: Isoquercitrin; AF: Afzelin.

**Figure 7 molecules-25-01920-f007:**
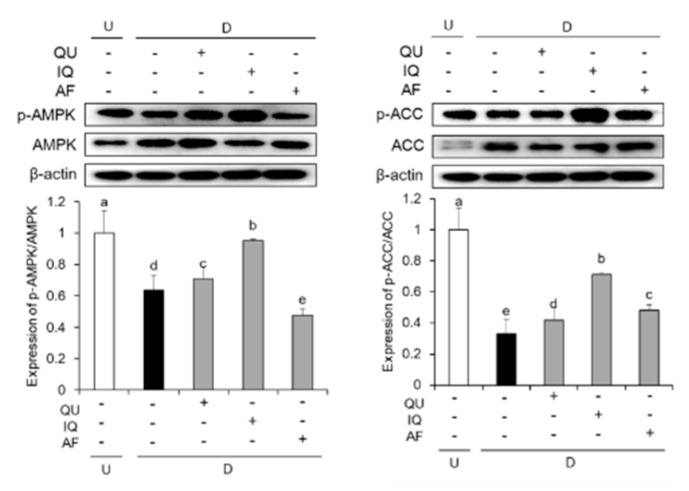
Effects of flavonoids from *Acer okamotoanum* (10 µg/mL) on AMPK signaling in differentiated 3T3-L1 cells. Adipocyte differentiation was induced by treatment with MDI media in the absence or presence of flavonoids from *A. okamotoanum* during 2 days. The MDI media was then replaced with insulin media, and it was changed four times for every 2 days. Data are expressed as the mean ± standard deviation. ^a^^–e^ Means with different letters indicate significant differences (*P* < 0.05) by Duncan’s multiple range test. β-actin was used as a loading control. β-actin was used as a loading control. Relative expression levels were normalized to the β-actin levels and the normal group. Normal group indicates non-differentiated cells, whereas control group indicates the differentiated cells by treatment of MDI media. U: Undifferentiation; D: Differentiation; QU: Quercitrin; IQ: Isoquercitrin; AF: Afzelin.

**Figure 8 molecules-25-01920-f008:**
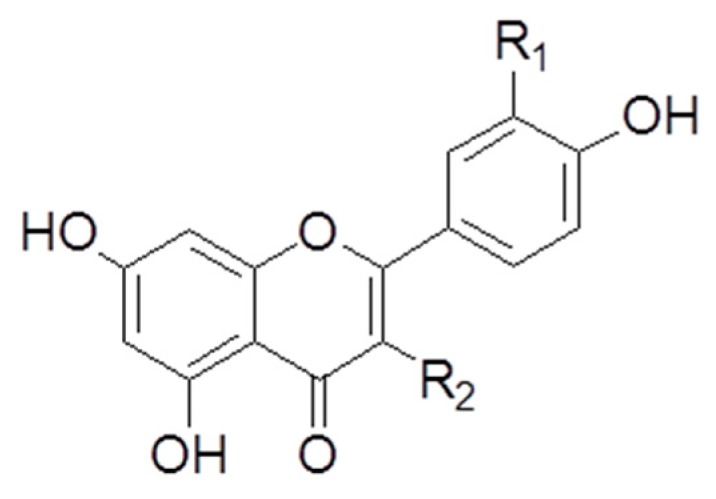
The structures of flavonoids from *Acer okamotoanum.*

**Table 1 molecules-25-01920-t001:** The Flavonoids from *Acer okamotoanum*.

Compound	R_1_	R_2_
QU	OH	*O*-Rhamnoside
IQ	OH	*O*-Glucoside
AF	H	*O*-Rhamnoside

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
