# Peer review of "Flavonoids from Acer okamotoanum Inhibit Adipocyte Differentiation and Promote Lipolysis in the 3T3-L1 Cells"

_molecules, 2020, doi:10.3390/molecules25081920_

Round 1

Reviewer 1 Report

This article exhibited the anti-obesity activities of 3 flavonoid compounds isolated from A.okamotoanum. According to the study in the differentiated 3T3-L1 cells, the authors deduced that flavonoids activate AMPK pathway.

Some questions are involved in the present study as following description.

  1. The regulatory effect of three flavonoids is inconsistent. Since the similar structure of flavonol glucosides, please discuss the possible action mechanism of different compounds. Besides the AMPK pathway, how about the roles of Wnt/β-catenin signaling pathway?
  2. When focused on the cell viability, the cytotoxicity of flavonoids treatment was not well stated. The cytotoxicity should be done in differentiated 3T3-L1 cells.

Author Response

Thank you for the valuable comments on this paper. We considered the comments carefully and the manuscript has been revised according to the comments.

This article exhibited the anti-obesity activities of 3 flavonoid compounds isolated from A.okamotoanum. According to the study in the differentiated 3T3-L1 cells, the authors deduced that flavonoids activate AMPK pathway.

Some questions are involved in the present study as following description.

1.The regulatory effect of three flavonoids is inconsistent. Since the similar structure of flavonol glucosides, please discuss the possible action mechanism of different compounds. Besides the AMPK pathway, how about the roles of Wnt/β-catenin signaling pathway?

; Thank you for the valuable comments. We discussed the possible mechanisms of various flavonoids including Wnt/β-catenin signaling pathway (Page 8-9, Line 275-296).

[Discussion]

Flavonoids are the most abundant polyphenols in natural products, and they can be classified into six groups including flavones, flavonols, flavanones, flavanonols, flavanols, and antocyanin (Panche et al., 2016). Dietary flavonoids from natural products exert anti-obesity effects via regulation of various molecular pathways in 3T3-L1 cells (Khalilpourfarshbafi et al., 2019). Flavones such as apigenin, balcalein, and lueteolin inhibited lipid accumulation by downregulation of adipogenesis, activation of AMPK, reduction of Akt-C/EBPa-GLUT4 signaling-medicated glucose uptake, and inflammatory responses (Ono and Fujimori, 2011; Nakao et al., 2016; Park et al., 2009). Flavanones including hesperitin and naringin exert antiobesity effects via activating PPAR and up-regulating adiponectin in 3T3-L1 cells (Liu et al., 2008). Anthocyanidins isolated from black soybean such as cyanidine, peonidin, and its glucoside reduced preadipocyte differentiation. Flavan-3-ols such as epigalloocatechin gallate in the green tea inhibited adipogenesis via regulation of Wnt/β-catenin pathway in 3T3-L1 cells (Khalilpourfarshbafi et al., 2019).

In our study, flavonoids from A. okamotoanum including QU, IQ, and AF are classified as flavonols. Flavonols including quercetin, kaempferol, and rutin are flavonoids with a ketone group in onions, tomatoes, apples, and berries (Panche et al., 2016). Flavonols such as quercetin, kaempferol, and its glycosides inhibited adipogenesis or promoted AMPK signaling in adipocytes (Lee et al., 2017; Torres-Villarreal et al., 2019). In comparison of aglycone and its glycosides, quercetin glucoside showed strong activity in reduction of lipid accumulation and inhibited adipogenic factors such as C/EBP-β, -α, and aP2 than its aglycone, quercetin (Lee et al., 2017). Wnt/β-catenin signaling is mandatory in adipogenesis, and it inhibited preadipocyte differentiation by regulation of β-catenin (Kennell and MacDougald, 2005). Lee et al. reported that IQ suppressed the adipogenesis in 3T3-L1 cells via the inhibition of Wnt/β-catenin signaling, regulation of lipid metabolism-related factors such as PPARγ, C/EBPα, SREBP-1, adiponectin, resistin, visfatin, and improvement of insulin resistance (Lee et al., 2011; Lee et al., 2017).

[References]

Kennell, J.A.; MacDougald, O.A. Wnt signaling inhibits adipogenesis through beta-catenin-dependent and -independent mechanisms. J. Biol. Chem. 2005, 280, 24004-24010.

Khalilpourfarshbafi, M.; Gholami, K.; Murugan, D. D.; Abdul, Sattar, M. Z.; Abdullah, N. A. Differential effects of dietary flavonoids on adipogenesis. Eur. J. Nutr. 2019, 58, 5-25.

Lee, C. W.; Seo, J. Y.; Lee, J.; Choi, J. W.; Cho, S.; Bae, J. Y.; Sohng, J. K.; Kim, S. O.; Kim, J.; Park, Y. I. 3-O-Glucosylation of quercetin enhances inhibitory effects on the adipocyte differentiation and lipogenesis. Biomed. Pharmacother. 2017, 95, 589-598.

Lee, S.H.; Kim, B.; Oh, M.J.; Yoon, J.; Kim, H.Y.; Lee, K.J.; Lee, J.D.; Choi, K.Y. Persicaria hydropiper (L.) spach and its flavonoid components, isoquercitrin and isorhamnetin, activate the Wnt/β-catenin pathway and inhibit adipocyte differentiation of 3T3-L1 cells. Phytother. Res. 2011, 25, 1629-1635.

Liu, L.; Shan, S.; Zhang, K.; Ning, Z.Q.; Lu, X.P.; Cheng, Y.Y. Naringenin and hesperetin, two flavonoids derived from Citrus aurantium up-regulate transcription of adiponectin. Phytother. Res. 2008, 22, 1400-1403.

Nakao, Y.; Yoshihara, H.; Fujimori, K. Suppression of very early stage of adipogenesis by baicalein, a plant-derived flavonoid through reduced Akt-C/EBPα-GLUT4 signaling-mediated glucose uptake in 3T3-L1 adipocytes. PLoS One 2016, 11, e0163640.

Ono, M.; Fujimori, K. Antiadipogenic effect of dietary apigenin through activation of AMPK in 3T3-L1 cells. J. Agric. Food Chem. 2011, 59, 13346-13352.

Panche, A.N.; Diwan, A.D.; Chandra, S.R. Flavonoids: an overview. J. Nutri. Sci. 2016, 5, e47.

Park, H.S.; Kim, S.H.; Kim, Y.S.; Ryu, S.Y.; Hwang, J.T.; Yang, H.J.; Kim, G.H.; Kwon, D.Y.; Kim, M.S. Luteolin inhibits adipogenic differentiation by regulating PPARgamma activation. Biofactors. 2009, 35, 373-379.

Torres-Villarreal, D.; Camacho, A.; Castro, H.; Ortiz-Lopez, R.; de la Garza, A. L. Anti-obesity effects of kaempferol by inhibiting adipogenesis and increasing lipolysis in 3T3-L1 cells. J. Physiol. Biochem. 2019, 75, 83-88.

2.When focused on the cell viability, the cytotoxicity of flavonoids treatment was not well stated. The cytotoxicity should be done in differentiated 3T3-L1 cells.

; We discussed it in discussion section (Page 7, Line 208-217).

[Discussion]

To determine the concentration of flavonoids, the cytotoxicity assay was carried out. Three flavonoids used in the present study had no significant cytotoxicity up to concentrations of 10 μg/mL (Figure 1). We previously investigated that three flavonoids had no cytotoxicity up to concentrations of 50 μg/mL in 3T3-L1 cells (data not shown). In addition, previous studies examined anti-adipogenic effects of IQ at concentrations of 5-100 μM in 3T3-L1 cells, indicating no cytotoxicity under higher concentrations used in this study (Cai et al., 2016; Lee et al., 2011). Therefore, in this study, inhibition of triglyceride accumulation by flavonoids was probably related to regulation of adipogenesis from 3T3-L1 preadipocytes to differentiated 3T3-L1 cells without cytotoxicity. In addition, we designed experimental schedule in measurement of cytotoxicity with reference to previous other studies (Hsu et al., 2010; Han et al., 2017).

[References]

Cai, H.D.; Su, S.L.; Guo, S.; Zhu, Y.; Qian, D.W.; Tao, W.W.; Duan, J.A. Effect of flavonoids from Abelmoschus manihot on proliferation, differentiation of 3T3-L1 preadipocyte and insulin resistance. Zhongguo Zhong Yao Za Zhi. 2016, 41, 4635-4641.

Lee, S.H.; Kim, B.; Oh, M.J.; Yoon, J.; Kim, H.Y.; Lee, K.J.; Lee, J.D.; Choi, K.Y. Persicaria hydropiper (L.) spach and its flavonoid components, isoquercitrin and isorhamnetin, activate the Wnt/β-catenin pathway and inhibit adipocyte differentiation of 3T3-L1 cells. Phytother. Res. 2011, 25, 1629-1635.

Hsu, H.F.; Tsou, T.C.; Chao, H.R.; Kuo, Y.T.; Tsai, F.Y.; Yeh, S.C. Effects of 2,3,7,8-tetrachlorodibenzo-p-dioxin on adipogenic differentiation and insulin-induced glucose uptake in 3T3-L1 cells. J. Hazard Mater. 2010, 182, 649-655.

Han, M.H.; Jeong, J.S.; Jeong, J.W.; Choi, S.H.; Kim, S.O.; Hong, S.H.; Park, C.; Kim, B.W.; Choi. Y.H. Ethanol extracts of Aster yomena (Kitam.) Honda inhibit adipogenesis through the activation of the AMPK signaling pathway in 3T3-L1 preadipocytes. Drug Discov. Ther. 2017, 11, 281-287.

Reviewer 2 Report

In the present report Kim et al investigated the effect of flavonoids from A. okamotoanum on 3T3L1 adipocytes. They found that these flavonoids reduced TG contents by inhibiting fatty acid synthesis and upregulated lipolysis related genes.

Comments

  1. In the toxicity/cell viability assay the authors should assess potential toxicity of concentrations higher than the one used in the experiments, reaching at least 10fold higher dose than the one suggested to have activity.

  1. It is not clear how long the treatment with the flavonoids lasted. This should be noted in all figure legends.

  1. In Figures 4-7 the terms ‘Normal’ and ‘Control’ are misleading; is should be noted as Undifferentiated and Differentiated.

  1. The fact that the flavonoids tested suppress the expression of GLUT4 it is possible that reduced TGs are due to reduced glucose uptake in the cells and reduced metabolism overall. This should be discussed.

  1. The authors should discuss their results in comparison with earlier studies on the effect of other flavonoids on 3T3L1 cells.

Author Response

Reviewer 2

Thank you for the valuable comments on this paper. We considered the comments carefully and the manuscript has been revised according to the comments.

In the present report Kim et al investigated the effect of flavonoids from A. okamotoanum on 3T3L1 adipocytes. They found that these flavonoids reduced TG contents by inhibiting fatty acid synthesis and upregulated lipolysis related genes.

Comments

1.In the toxicity/cell viability assay the authors should assess potential toxicity of concentrations higher than the one used in the experiments, reaching at least 10fold higher dose than the one suggested to have activity.

; We discussed it in discussion section (Page 7, Line 208-217).

[Discussion]

To determine the concentration of flavonoids, the cytotoxicity assay was carried out. Three flavonoids used in the present study had no significant cytotoxicity up to concentrations of 10 μg/mL (Figure 1). We previously investigated that three flavonoids had no cytotoxicity up to concentrations of 50 μg/mL in 3T3-L1 cells (data not shown). In addition, previous studies examined anti-adipogenic effects of IQ at concentrations of 5-100 μM in 3T3-L1 cells, indicating no cytotoxicity under higher concentrations used in this study (Cai et al., 2016; Lee et al., 2011). Therefore, in this study, inhibition of triglyceride accumulation by flavonoids was probably related to regulation of adipogenesis from 3T3-L1 preadipocytes to differentiated 3T3-L1 cells without cytotoxicity. In addition, we designed experimental schedule in measurement of cytotoxicity with reference to previous other studies (Hsu et al., 2010; Han et al., 2017).

[References]

Cai, H.D.; Su, S.L.; Guo, S.; Zhu, Y.; Qian, D.W.; Tao, W.W.; Duan, J.A. Effect of flavonoids from Abelmoschus manihot on proliferation, differentiation of 3T3-L1 preadipocyte and insulin resistance. Zhongguo Zhong Yao Za Zhi. 2016, 41, 4635-4641.

Lee, S.H.; Kim, B.; Oh, M.J.; Yoon, J.; Kim, H.Y.; Lee, K.J.; Lee, J.D.; Choi, K.Y. Persicaria hydropiper (L.) spach and its flavonoid components, isoquercitrin and isorhamnetin, activate the Wnt/β-catenin pathway and inhibit adipocyte differentiation of 3T3-L1 cells. Phytother. Res. 2011, 25, 1629-1635.

Hsu, H.F.; Tsou, T.C.; Chao, H.R.; Kuo, Y.T.; Tsai, F.Y.; Yeh, S.C. Effects of 2,3,7,8-tetrachlorodibenzo-p-dioxin on adipogenic differentiation and insulin-induced glucose uptake in 3T3-L1 cells. J. Hazard Mater. 2010, 182, 649-655.

Han, M.H.; Jeong, J.S.; Jeong, J.W.; Choi, S.H.; Kim, S.O.; Hong, S.H.; Park, C.; Kim, B.W.; Choi. Y.H. Ethanol extracts of Aster yomena (Kitam.) Honda inhibit adipogenesis through the activation of the AMPK signaling pathway in 3T3-L1 preadipocytes. Drug Discov. Ther. 2017, 11, 281-287.

2.It is not clear how long the treatment with the flavonoids lasted. This should be noted in all figure legends.

; We added it in Figure legends.

[Figure legends]

The 3T3-L1 adipocytes were pretreated with various concentrations (1–10 µg/mL) of flavonoids from A. okamotoanum for 72 h.

Adipocyte differentiation was induced by treatment with MDI media in the absence or presence of flavonoids from A. okamotoanum during 2 days. The MDI media was then replaced with insulin media, and it was changed four times for every 2 days.

3.In Figures 4-7 the terms ‘Normal’ and ‘Control’ are misleading; is should be noted as Undifferentiated and Differentiated.

; We revised it in Figure 2-7.

4.The fact that the flavonoids tested suppress the expression of GLUT4 it is possible that reduced TGs are due to reduced glucose uptake in the cells and reduced metabolism overall. This should be discussed.

; According to the comment, it was discussed. We discussed it in discussion section (Page 8, Line 239-245).

[Discussion]

Expression of GLUT4 is increased by activation of adipogenic key transcription factor such as C/EBPα in adipocyte differentiation (Kaestner et al., 1990). Previous study reported that decrease of C/EBPα inhibited expression of GLUT, which would decrease glucose transport in adipocytes (Kaestner et al., 1990; Watanabe et al., 2015). In our study, treatment of IQ decreased the expression of C/EBPα and GLUT4. Therefore, IQ inhibited the accumulation of TGs by decreasing C/EBPα-activated GLUT4 expression. GLUT4 is a member of glucose transporter in adipose tissue, and involved in the insulin-stimulated glucose uptake (Abel et al., 2001; Qiu et al., 2012). On the basis of these evidences, the inhibitory effect of IQ on TG accumulation is related to GLUT4 by regulation of C/EBPα expressions.

[References]

Kaestner K.H., Christy R.J., Lane M.D. Mouse insulin-responsive glucose transporter gene: Characterization of the gene and trans-activation by the CCAAT/enhancer binding protein. Proc. Natl. Acad. Sci. USA. 1990, 87, 251–255.

Watanabe, M.; Hisatake, M.; Fujimori, K. Fisetin suppresses lipid accumulation in mouse adipocytic 3T3-L1 cells by repressing GLUT4-mediated glucose uptake through inhibition of mTOR-C/EBPα signaling. J. Agric. Food Chem. 2015, 63, 4979–4987.

Abel, E.D.; Peroni, O.; Kim, J.K.; Kim, Y.B.; Boss, O.; Hadro, E.; Minnemann T.; Shulman, G.I.; Kahn, B.B. Adipose-selective targeting of the GLUT4 gene impairs insulin action in muscle and liver. Nature 2001, 409, 729–733.

Qiu, J.; Wang, Y.M.; Shi, C.M.; Yue, H.N.; Qin, Z.Y.; Zhu, G.Z.; Cao, X.G.; Ji, C.B.; Cui, Y.; Guo, X.R. NYGGF4 (PID1) effects on insulin resistance are reversed by metformin in 3T3-L1 adipocytes. J. Bioenerg. Biomembr. 2012, 44, 665-671.

5.The authors should discuss their results in comparison with earlier studies on the effect of other flavonoids on 3T3L1 cells.

; Thank you for the valuable comments. We discussed the possible mechanisms of various flavonoids including Wnt/β-catenin signaling pathway (Page 8-9, Line 275-296).

[Discussion]

Flavonoids are the most abundant polyphenols in natural products, and they can be classified into six groups including flavones, flavonols, flavanones, flavanonols, flavanols, and antocyanin (Panche et al., 2016). Dietary flavonoids from natural products exert anti-obesity effects via regulation of various molecular pathways in 3T3-L1 cells (Khalilpourfarshbafi et al., 2019). Flavones such as apigenin, balcalein, and lueteolin inhibited lipid accumulation by downregulation of adipogenesis, activation of AMPK, reduction of Akt-C/EBPa-GLUT4 signaling-medicated glucose uptake, and inflammatory responses (Ono and Fujimori, 2011; Nakao et al., 2016; Park et al., 2009). Flavanones including hesperitin and naringin exert antiobesity effects via activating PPAR and up-regulating adiponectin in 3T3-L1 cells (Liu et al., 2008). Anthocyanidins isolated from black soybean such as cyanidine, peonidin, and its glucoside reduced preadipocyte differentiation. Flavan-3-ols such as epigalloocatechin gallate in the green tea inhibited adipogenesis via regulation of Wnt/β-catenin pathway in 3T3-L1 cells (Khalilpourfarshbafi et al., 2019).

In our study, flavonoids from A. okamotoanum including QU, IQ, and AF are classified as flavonols. Flavonols including quercetin, kaempferol, and rutin are flavonoids with a ketone group in onions, tomatoes, apples, and berries (Panche et al., 2016). Flavonols such as quercetin, kaempferol, and its glycosides inhibited adipogenesis or promoted AMPK signaling in adipocytes (Lee et al., 2017; Torres-Villarreal et al., 2019). In comparison of aglycone and its glycosides, quercetin glucoside showed strong activity in reduction of lipid accumulation and inhibited adipogenic factors such as C/EBP-β, -α, and aP2 than its aglycone, quercetin (Lee et al., 2017). Wnt/β-catenin signaling is mandatory in adipogenesis, and it inhibited preadipocyte differentiation by regulation of β-catenin (Kennell and MacDougald, 2005). Lee et al. reported that IQ suppressed the adipogenesis in 3T3-L1 cells via the inhibition of Wnt/β-catenin signaling, regulation of lipid metabolism-related factors such as PPARγ, C/EBPα, SREBP-1, adiponectin, resistin, visfatin, and improvement of insulin resistance (Lee et al., 2011; Lee et al., 2017).

[References]

Kennell, J.A.; MacDougald, O.A. Wnt signaling inhibits adipogenesis through beta-catenin-dependent and -independent mechanisms. J. Biol. Chem. 2005, 280, 24004-24010.

Khalilpourfarshbafi, M.; Gholami, K.; Murugan, D. D.; Abdul, Sattar, M. Z.; Abdullah, N. A. Differential effects of dietary flavonoids on adipogenesis. Eur. J. Nutr. 2019, 58, 5-25.

Lee, C. W.; Seo, J. Y.; Lee, J.; Choi, J. W.; Cho, S.; Bae, J. Y.; Sohng, J. K.; Kim, S. O.; Kim, J.; Park, Y. I. 3-O-Glucosylation of quercetin enhances inhibitory effects on the adipocyte differentiation and lipogenesis. Biomed. Pharmacother. 2017, 95, 589-598.

Lee, S.H.; Kim, B.; Oh, M.J.; Yoon, J.; Kim, H.Y.; Lee, K.J.; Lee, J.D.; Choi, K.Y. Persicaria hydropiper (L.) spach and its flavonoid components, isoquercitrin and isorhamnetin, activate the Wnt/β-catenin pathway and inhibit adipocyte differentiation of 3T3-L1 cells. Phytother. Res. 2011, 25, 1629-1635.

Liu, L.; Shan, S.; Zhang, K.; Ning, Z.Q.; Lu, X.P.; Cheng, Y.Y. Naringenin and hesperetin, two flavonoids derived from Citrus aurantium up-regulate transcription of adiponectin. Phytother. Res. 2008, 22, 1400-1403.

Nakao, Y.; Yoshihara, H.; Fujimori, K. Suppression of very early stage of adipogenesis by baicalein, a plant-derived flavonoid through reduced Akt-C/EBPα-GLUT4 signaling-mediated glucose uptake in 3T3-L1 adipocytes. PLoS One 2016, 11, e0163640.

Ono, M.; Fujimori, K. Antiadipogenic effect of dietary apigenin through activation of AMPK in 3T3-L1 cells. J. Agric. Food Chem. 2011, 59, 13346-13352.

Panche, A.N.; Diwan, A.D.; Chandra, S.R. Flavonoids: an overview. J. Nutri. Sci. 2016, 5, e47.

Park, H.S.; Kim, S.H.; Kim, Y.S.; Ryu, S.Y.; Hwang, J.T.; Yang, H.J.; Kim, G.H.; Kwon, D.Y.; Kim, M.S. Luteolin inhibits adipogenic differentiation by regulating PPARgamma activation. Biofactors. 2009, 35, 373-379.

Torres-Villarreal, D.; Camacho, A.; Castro, H.; Ortiz-Lopez, R.; de la Garza, A. L. Anti-obesity effects of kaempferol by inhibiting adipogenesis and increasing lipolysis in 3T3-L1 cells. J. Physiol. Biochem. 2019, 75, 83-88.